# Nominal Groups to Develop a Mobile Application on Healthy Habits

**DOI:** 10.3390/healthcare9040378

**Published:** 2021-03-30

**Authors:** Mª Soledad Palacios-Gálvez, Montserrat Andrés-Villas, Mercedes Vélez-Toral, Ángeles Merino-Godoy

**Affiliations:** 1Department of Social, Developmental and Educational Psychology, University of Huelva, 21007 Huelva, Spain; maria.palacios@dpsi.uhu.es (M.S.P.-G.); montserrat.andres@dpsi.uhu.es (M.A.-V.); maria.velez@dpee.uhu.es (M.V.-T.); 2Center for the Investigation of Contemporary Thought and Innovation for Social Development (COIDESO), University of Huelva, 21007 Huelva, Spain; 3Department of Nursing, University of Huelva, 21007 Huelva, Spain

**Keywords:** education, health, adolescence, mobile application, nominal groups

## Abstract

The new pandemic-lockdown situation has caused empowerment of new technologies; mobile phones and computers have gained further importance. Homes have become the new educators of health since health education has decreased or stopped during the pandemic. The lack of knowledge in the child and adolescent population about how to incorporate healthy habits in their daily lives, along with the rise of health devices and the introduction of healthcare in the syllabus, has led to the realization of the present study. The aim of this study was to identify the relevant health topics in a sample of adolescents for the later development of a mobile application (Healthy Jeart) that promotes the adoption of healthy lifestyle habits in adolescence. The information was gathered through the technique of nominal groups. The sample was recruited by nonprobability purposive sampling, with a total of 92 students from 4 educational centers of the province of Huelva (Spain). According to the obtained results, the most relevant categories were physical wellbeing (40.81%), psychological wellbeing (22.13%), interpersonal relationships and social skills (21.58%), toxic substances and addictions (10.35%) and sex habits (1.83%). This technique allowed identifying and selecting the most relevant content areas of the “Healthy Jeart” application.

## 1. Introduction

According to the World Health Organization [1], among other characteristics, the services that are aimed at improving the health of adolescents must be accessible, acceptable and appropriate, and they must meet their expectations and needs. As was stated by Gray et al. [2], adolescents find it difficult to access information about health through traditional methods, and the Internet is part of their world and a confidential, accessible and quick way of obtaining information. A study with young Andalusians showed that 55.7% of the surveyed sample used the Internet to obtain information about topics related to health [3]. The Internet and mobile devices are very powerful instruments to provide information and influence the attitudes, values and behaviors that strengthen adolescents’ health, and they are part of mHealth or eHealth (health care supported by information and communication technologies) [4].

The global pandemic caused by the new COVID-19 has generated stress worldwide, both in the health system and throughout society. In Spain, it has meant a radical change in practicing family and community nursing since education and health promotion have been affected since consultations with education for chronic or healthy children have been paralyzed or canceled. For this reason, we have to consider what changes are currently necessary for the health system to provide quality care in which measures for the development of prevention and health promotion are implemented [5].

Obesity, diabetes, high blood pressure and smoking have been manifested as aggravating factors in COVID-19 infection; therefore, investment in prevention and health promotion is also essential to combat this pandemic [5] and its consequences, since one out of four children who have suffered the lockdown has anxiety and depressive symptoms [6]. This situation could affect healthy lifestyles and negatively impact physical and mental health [7]. Studies suggest that, in children, non-school periods are associated with lower physical activity, irregular sleep patterns, and less healthy diets [8]. In a lockdown situation, these unhealthy habits increase due to added circumstances, such as having outdoor activities restricted (with an increased sedentary behavior) [9]. Therefore, it has been necessary to implement telephone assistance, telemedicine and videoconferencing as a substitute for face-to-face assistance. Hence, the present time is largely characterized by online teaching and teleworking.

Due to the COVID-19 pandemic, the population has suffered home confinement worldwide; thus, in order to continue with our lives, we had to resort to new technologies, with mobile phones and computers being the great protagonists, for both children and adults. Homes have become the new health educators since, also as a consequence of this situation; health education has decreased or stopped. Therefore, through this project, we aim to bring health education to adolescents, as numerous studies have confirmed the impact of eHealth on health.

Currently, schools are introducing health care in their syllabi. Therefore, to promote healthy behaviors in adolescents, the mobile application (app) “Healthy Jeart” was created and is made up of four different and interconnected sections: a game, a forum of ideas, health advice (tips) and challenges. The development process and the App’s description have been previously presented in detail by Palacios-Gálvez, Yot-Domínguez and Merino-Godoy [10], and Duarte-Hueros, Yot-Domínguez and Merino-Godoy [11], published in the Spanish Journal of Public Health and in Education and Information Technologies. One of these studies described the entire design process (Figure 1), whereas the other one gathers a thorough description of the tool. Also, there is a website associated with the App (www.healthyjeart.com, accessed on 26 February 2021), where you can find information about the App and also activities and other resources for teachers about different health area.

The aim of the present study was to identify the health-related needs and concerns of a sample of children and adolescents to be used for the later development of the app.

## 2. Methods

To begin the creation of the tool, and following the recommendations of the WHO [12] regarding the fact that “adolescents must be heard and participate in the planning, execution, followup and evaluation of new services”, the technique of nominal groups (NG), originally developed by Delbecq and Van de Ven [13], was used to explore the health knowledge-related needs and concerns in the pre- and adolescent population from a bio-psycho-social approach, in order to select the type of contents that would be included in the app.

NG is a useful strategy to obtain information in a participatory and structured manner, identify problems and establish solutions and priorities. Ideas are generated in a relaxed environment, in which the participants share their thoughts, orally or in writing. This technique allowed reaching balanced participation among all participants, which is more difficult to achieve with other techniques, such as discussion groups or the Delphi method [14].

The sample was recruited by nonprobability purposive sampling, with a total of 92 students from 4 educational centers of the province of Huelva (Spain): 26 from year 6 (11–12 years old), 28 from year 9 (15–16 years old), 29 from year 10 (16–17 years old) and 9 from the University of Huelva (Spain) (18–22 years old). These are ages of great changes; thus, it was necessary to recruit children and adolescents of different ages. In all cases, informed consent was obtained (from their parents/legal guardians for those under age), and the study was approved by the Research Ethics Committee.

In the first contact with the principals and directors of the centers, to show them the project, we had a positive, participatory response since they all agreed on three aspects:

First, they agreed on the need to approach healthy habits due to the increasing deterioration of the lifestyle in these ages: a diet based mostly on processed foods, sedentary lifestyle, poor sleep hygiene, etc.

Second, they thought it was important to know the interests of their students about their health habits.

Third, they considered that a didactic tool based on new technologies could be very helpful for working on these topics in the classroom in an attractive manner.

Eight nominal groups (3 with year 6 students, 2 with year 9 students, 2 with year 10 students and 1 with university students) were carried out in the corresponding educational centers during school hours. The mean number of participants in each group was 11.5 students, with a maximum of 15 and a minimum of 8. In each session, after giving the participants information about the project and the purpose of the app, they were asked to individually write down the health topics they thought were most interesting to be included in the app. Then, all the answers were put together and grouped based on similarity; subsequently, the participants were asked to choose 5 of those proposals and order them by scoring them according to their own preference (between 5 points, for the most interesting idea, and 1 point, for the least interesting one). The session concluded with a brief presentation of the results.

## 3. Results

Table 1 shows the results of the most voted proposals, organized in seven general categories and classified into 23 subcategories, all of which were created a posteriori by the research group. The scores obtained by each of them were summed (adding the points assigned), and the percentages were calculated over the total, with the aim of generating a preference order.

As can be observed in Table 1, the most relevant category is physical wellbeing (with slightly over 40%), followed by psychological wellbeing and social relationships, which are very close to each other (22.13% and 21.58%, respectively). The fourth category is the one related to toxic substances and addictions (10.35%), followed by the “miscellaneous” category (2.91%), sex habits (1.83%) and the “external” or structural factors that influence health (poverty) (0.39%). The latter case is a factor proposed by a student of primary education (year 6), whose obtained votes are the ones that she gave it since no other response was included in that category.

Analyzing the results by academic level, Table 2 shows the clear preference of the primary education students for the information related to physical wellbeing, followed by toxic substances and addictions. Regarding the secondary education students (years 9 and 10), there was also a clear inclination for the topics related to physical wellbeing, followed by psychological wellbeing and social relationships, with the latter two being very close to each other.

The year 10 students had a clear preference for psychological wellbeing, followed by physical wellbeing and social relationships. They showed a slightly greater interest in toxic substances and addictions with respect to sex habits.

Lastly, the university students were more interested in social relationships and physical wellbeing, followed by psychological wellbeing. Similarly, and in line with the results obtained by the other groups, their interest in sex habits was lower, showing a greater interest in toxic substances and addictions. These results allowed establishing a diagnosis of the health education needs at this stage; thus, the topics highlighted by the participants are included in the design of all the sections of this digital health tool.

## 4. Discussion

Adolescence is an important stage for the promotion of present and future health. As has been pointed out by the WHO [12], health interventions must stop focusing solely on the prevention of pregnancy and HIV transmission and should expand their scope in order to attend to the more specific needs of this group. The aim of this study was to identify these needs and concerns of young people using nominal groups (NG), which allowed establishing relevant topics and organizing them based on the importance given to them by youths of different age groups. As limitations of the study, other moderator variables have not been included, such as sex, intelligence quotient, levels of mental health, and suffering from specific health conditions, such as diabetes. These variables could actually spring further insights on the matter and entail new studies”.

The results showed that the general preferences of the four groups are focused, according to priority, on physical wellbeing (diet and physical care, such as hygiene and rest) and physical activity, psychological wellbeing, interpersonal relationships and, lastly, toxic substances and addictions. Although initially, only the most voted categories and subcategories would be included as part of the app’s content, eventually, it was decided to include sex habits as well.

In view of these results, NG seems to be a useful tool to clearly identify the relevant health-related topics that concern young people the most and, therefore, allows detecting action priorities linked to the motivations of this specific population and associated with the age and stage of development. In this case, the implementation of this technique allowed identifying and selecting the content areas of the “Healthy Jeart” app for the promotion of health in adolescence in a fun and simple way with great scientific rigor (Figure 2). As was previously mentioned, home confinement due to the COVID-19 pandemic has led new technologies to become the protagonists of our daily life. Moreover, as a result of the work overload of health centers, health education has been reduced or even stopped; therefore, Healthy Jeart allows resuming health education and bringing it to children and adolescents in an innovative manner. In 2018, this app was recognized as an “AppSaludable” (Spanish distinction, which means “Healthy app”) by the Andalusian Agency of Health Quality (ACSA).

As was stated by Aguilera, Ruíz and Utrera [15], healthy apps and websites have a great impact on the population at the bio-psycho-social level, since they are an important source of motivation and a new way to access the children and adolescent population, which allows improving the capacity to receive and transmit information to the users. Therefore, we must ensure that their content is of sufficient relevance and novelty to reach our population and thus orient the teaching of the users based on their education level. In response to the research question (“Can mobile applications transform health care?”), new technologies can help people to reduce morbidity and mortality through the promotion of healthy behaviors [15].

To sum up, the technique of nominal groups enabled us to identify the health-related needs and concerns of a sample of adolescents. In this line, the authors of the contents of Healthy Jeart are experts in the thematic area they are in charge of. Once they gather the updated content, and according to the scientific evidence, they send the material of their field of knowledge to a national expert, who presents the necessary modifications before such content is published in the app. Likewise, the content is updated annually by the authors of each area. This research project was the foundation for the improvement of a low-cost health prevention and promotion tool and a guide for the development of the questionnaires. In this sense, in a new research phase, we will validate the questionnaires designed according to the content of healthy habits of this tool, which will allow measuring the impact of the use of this application (Healthy Jeart) on health.

## Figures and Tables

**Figure 1 healthcare-09-00378-f001:**
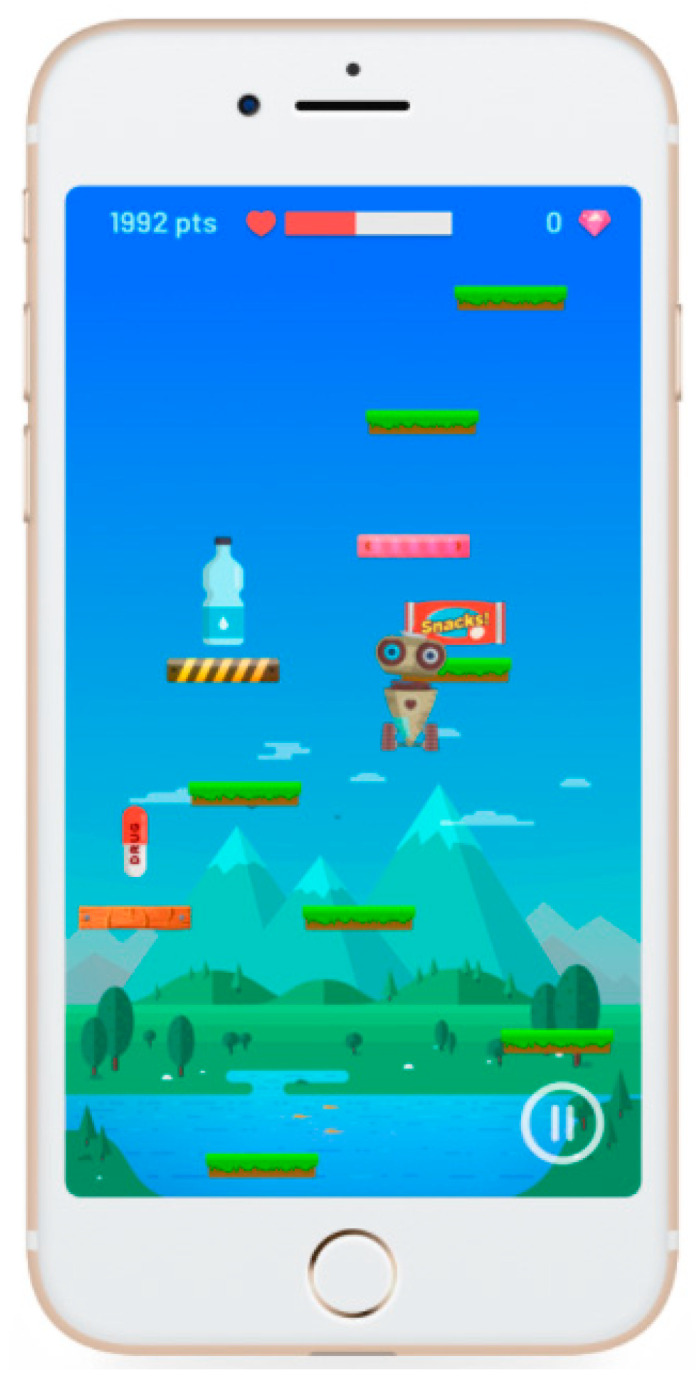
Image of the game (from the website: www.healthyjeart.com, accessed on 26 February 2021).

**Figure 2 healthcare-09-00378-f002:**
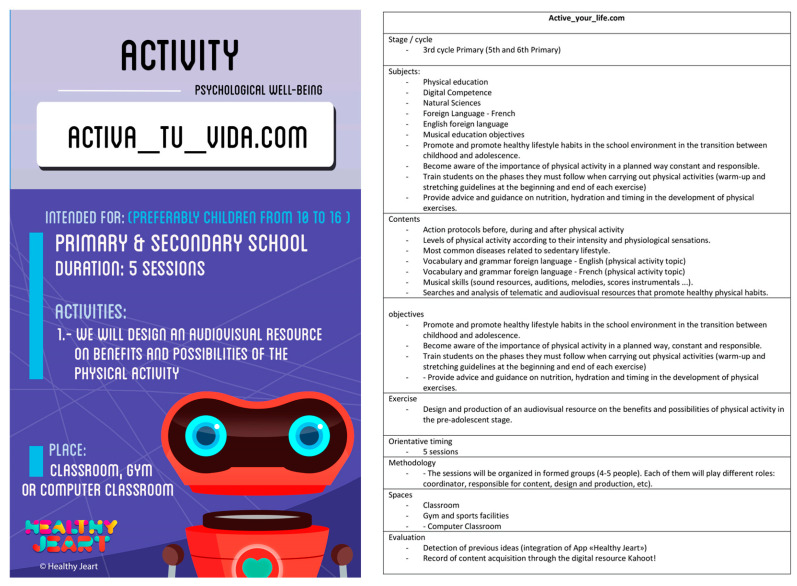
Image from the website. Frontal page of the didactic activities of physical activity.

**Table 1 healthcare-09-00378-t001:** Categories, subcategories and descriptors. Percentages of their obtained points and order of preference (calculated from the percentages).

Categories % of Points over the Total (Position)	Subcategories % of Points over the Total (Position)	Descriptors
Physical wellbeing 40.81% (1°)	Physical activity/9.03% (4°)	Exercise, sport
	Diet/14.61% (2°)	Eating habits, diet, hydration
	Cleanliness and hygiene/4.51% (8°)	Personal hygiene, room tidiness, clothing, etc.
	Environment/1.20% (15°)	Habits to protect the environment
	Rest/4.16% (9°)	Habits related to basic rest
	Physical care/7.31% (6°)	Prevention and protection behaviors: (e.g., skin, lungs, inner ears, vaccination, body posture, etc.); home remedies; road safety, etc.
Psychological wellbeing 22.13% (2°)	Studies/1.05% (16°)	Study habits; concentration
	Time management/1.52% (14°)	Time administration/organization
	Healthy leisure/1.99% (12°)	Listening to music, reading, singing, dancing, doing things one likes, traveling, going to cultural events
	Feeling well with oneself/16.95% (1°)	Self-esteem, stress management, emotional balance.
	Social support/0.57% (19°)	Search for help in the face of a problem (e.g., bullying)
	Autonomy/0.62% (18°)	Reflecting, internalizing to know oneself, mental exercise, spending time in hobbies, setting goals and objectives, relaxation and meditation
Interpersonal relationships and social skills 21.58% (3°)	Social relationships/12.33% (3°)	Social skills
	Social relationships through ICT/0.33% (22°) Prosocial behaviors/7.00% (7°)	Safety in social relationships Promotion of respect, solidarity and tolerance. Avoiding discrimination, etc.
	Relationships with reference adults/0.45% (20°)	Listening to older people
	Relationships with friends/0.91% (17°)	Preventing harassment among peers (bullying)
	Social relationships/12.33% (3°)	Social skills
Toxic substances and addictions 10.35% (4°)	Abuse of new technologies/2.15% (11°)	Use/abuse of new technologies
	Alcohol, drugs, smoking/8.20% (5°)	Information/prevention of the consumption of alcohol, cigarettes, shisha, other drugs, etc.
Miscellaneous 2.91% (5°)	Habits related to uncategorized topics/2.92% (10°)	Suicide; death; emergency number, helplines.
Sex habits 1.83% (6°)	Sex habits/1.83% (12°)	STDs, affective sexual education
Other factors that influence health 0.39% (7°)	Sociostructural factors/0.39% (21°)	Poverty
TOTAL: 100%	Total: 100%	

Source: developed by the author.

**Table 2 healthcare-09-00378-t002:** Percentages of the points obtained by each category, according to the academic groups.

	Year 6	Year 9	Year 10	University
Physical wellbeing	54.16%	56.31%	20.92%	31.85%
Psychological wellbeing	7.81%	20.56%	40.88%	19.26%
Social relationships	15.36%	15.69%	18.98%	36.3%
Toxic substances and addictions	18.75%	6.94%	7.54%	8.14%
Sex habits	0%	0%	5.11%	2.22%
Miscellaneous	2.86%	0%	6.57%	2.22%
Other factors	1.04%	0.51%	0%	0%
Total	100%	100%	100%	100%

Source: developed by the author.

## Data Availability

Not applicable.

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
