# Peer review of "Nominal Groups to Develop a Mobile Application on Healthy Habits"

_healthcare, 2021, doi:10.3390/healthcare9040378_

Round 1
Reviewer 1 Report
Thank you for the opportunity to review your manuscript describing the use of nominal groups to identify important health elements for pre-and adolescents. Your methods were well described and methodologically sound. Just a few suggestions for further clarification of your process/findings:
- It would be helpful to see the basic demographics of the 92 participants (gender, race). Was there any difference in responses by gender or race?
- Did any of the participants have any diagnosed health conditions such as Type I diabetes, depression, etc.? These responses may be different from participants that did not have any diagnosed health conditions. It would be helpful to see if you have a representation from participants with a diagnosed health condition and those without.
- Unclear as to what exactly is in the app. You mention a game on page 6. Is that how the health information was delivered? Need details on the app itself.
- What does Figure 2 (website) have to do with the app?
- On page 6 you discuss the pilot testing of the app. How many participants? Basic demographics? How long? How did you collect the feedback?
- On page 6 you stated the gamification piece did not appeal to those over age 16 and the app went through a series of revisions. It is unclear how you are delivering health information.
- I would suggest you remove the information about the testing of the app. You do not provide enough information for the reader to completely understand the studies.
- Last paragraph on page 7. You mention health apps and websites, which links back to question #3. Is this an app or a website the participants can access via mobile phone?
- The title suggests that you are only describing the use of the nominal groups. However, you have included other details such as pilot testing and 2 additional studies. I am assuming you are writing additional manuscripts from these steps; therefore, I would suggest that you only focus on the nominal group piece in this manuscript.
Reviewer 2 Report
The work entitled “Nominal groups to develop a mobile application on healthy habits” contains new scientific knowledge and covers a relevant topic. However, I have some comments that have to be addressed before it can be considered for publication.
In my opinion, the introduction should provide a more detailed analysis about previous research on adolescent mental health and addictions.
I think that authors must explain this affirmation: “These are ages of great changes, thus it was necessary to recruit students of different educational stages”. I do not understand why it is necessary to recruit students because of the age.
I would also suggest adding more information about the participants subsection. For instance mean age and standard deviation, age distribution, gender distribution. If authors considered previous or actual history of mental health, IQ or another measure of intelligence, or another relevant variable. Otherwise, this could be mentioned in the limitations subsection.
In the discussion, I would suggest authors modifying this affirmation: “…of evolutionary development”. It is more appropriate to talk about development than evolution (this term is related to species not the individual person).
